# Three Cases of Atypical Pneumonia with *Chlamydia psittaci*: The Role of Laboratory Vigilance in the Diagnosis of Psittacosis

**DOI:** 10.3390/pathogens12010065

**Published:** 2022-12-31

**Authors:** Sophie Missault, Anne De Meyst, Jan Van Elslande, Anne-Marie Van den Abeele, Elke Steen, Jos Van Acker, Daisy Vanrompay

**Affiliations:** 1Department of Geriatric Medicine, AZ Sint-Lucas Hospital, 9000 Ghent, Belgium; 2Laboratory for Immunology and Animal Biotechnology, Department of Animal Science and Aquatic Ecology, Ghent University, 9000 Ghent, Belgium; 3Laboratory of Clinical Microbiology, AZ Sint-Lucas Hospital, 9000 Ghent, Belgium

**Keywords:** psittacosis, *Chlamydia psittaci*, atypical pneumonia, zoonosis

## Abstract

*Chlamydia psittaci* is an established zoonotic agent causing respiratory disease in humans. An infection often remains asymptomatic but can also result in flu-like illness, pneumonia or even multi-organ failure. This paper describes three patients, hospitalised at AZ Sint-Lucas Hospital, with atypical pneumonia who were diagnosed with *C. psittaci* after an in-depth anamnesis and laboratory investigation in the midst of the COVID pandemic. All three infections were confirmed with PCR and serology, whereas viable bacteria were only present for one patient. Genotyping revealed the presence of genotype B for patient 1 and 2 whereas *ompA* genotyping was unsuccessful for patient 3. This case report demonstrates the importance of a thorough patient history as close contact with birds is one of the main risk factors to contract the pathogen. Once exposure to birds has been confirmed, a diagnosis by a combination of PCR and serology is essential in order to initiate a treatment with the proper antibiotics. As psittacosis is still an underestimated and underdiagnosed disease, communication between laboratory, clinicians and bird fanciers is encouraged.

## 1. Introduction

*Chlamydia psittaci* is the causative agent of psittacosis, also known as parrot fever. It is an obligate intracellular bacterium that belongs to the *Chlamydiaceae* family, which currently comprises 14 characterized species [1,2].

*C. psittaci* is an established zoonotic pathogen, infecting humans predominantly after contact with infected birds or their excrements. The typical reservoir concerns parrot-type birds (*Psittaciformes)*, although many other birds (e.g., pigeons, turkeys, ducks, chickens, owls) and even mammals such as horses have been implicated in zoonotic transmission of *C. psittaci* to humans [3]. Human-to-human transmission is possible but rarely occurs [4,5]. As *C. psittaci* is mainly transmitted through aerosols, individuals having regular close contact with birds are an important risk population.

Human infections occur as solitary events or in outbreak settings, such as in poultry processing plants, poultry farms, bird shows and bird sanctuaries. Especially during outbreaks, laboratory diagnostics are indispensable to aid with case finding, cluster identification and source detection [6].

Symptoms of psittacosis range from none (asymptomatic) to mild flu-like illness up to severe pneumonia. Sepsis, hepatitis and/or meningitis can occur when the disease becomes systemic, which is sporadically fatal [7,8,9]. As there are no specific symptoms of psittacosis, compared to other atypical pneumonias, the incidence of psittacosis is largely underestimated [1].

Psittacosis is a mandatory, notifiable disease in all three Belgian regions (Brussels, Flanders and Wallonia). In general, a case can be confirmed when the patient has clinical signs like headache, cough, fever and chills, in combination with strong laboratory evidence provided by isolation of *C. psittaci* from respiratory secretions (preferably samples of the lower respiratory tract) or blood, or by a significant antibody titer rise on paired acute and convalescent serum samples. A probable case implies clinical signs and a single increase in *C. psittaci*-specific IgM antibodies (Wallonia or Flanders) or the detection of *C. psittaci* in respiratory secretions by PCR (Flanders) [1].

We describe three cases of pneumonia in patients with a history of close contact with pigeons and parrots during the COVID pandemic. All three patients were diagnosed with psittacosis after in-depth laboratory testing. This study demonstrates the importance of a thorough anamnesis in combination with laboratory vigilance for rapid case confirmation.

## 2. Case Description

### 2.1. Patient 1

An 87-year-old man was referred to the emergency care department by his general care practitioner because of progressive dyspnea, fever and general malaise for one week. The patient was empirically prescribed azithromycin, with insufficient improvement. The man had a history of atrial flutter and coronary artery disease and mentioned that he was a pigeon owner. At the time he was double vaccinated against COVID-19 and SARS-CoV-2 PCR-negative. Laboratory investigation showed an acute kidney failure and moderately elevated inflammatory parameters with a CRP of 47 mg/L (reference value <5 mg/L) and a leukocytosis of 13.7 × 10^3^/μL (reference value 3.4–9.8 × 10^3^/μL) in presence of a left shift. A chest CT scan revealed bilateral patchy infiltrates and lung emphysema. The patient was hospitalized and treated empirically for atypical pneumonia with moxifloxacin for ten days. Clinical symptoms improved and the patient could be discharged after eight days. One week after dismissal, *C. psittaci*-specific PCR on sputum, performed by the national reference laboratory for human psittacosis (Ghent University), returned positive. Serology for *C. psittaci* was significantly positive as well, and in consultation with the treating pneumologist, the patient was treated with doxycycline (100 mg once daily after a loading dose of 200 mg) for another ten days. A re-evaluation after three months revealed a complete resolution of his pneumonia on chest X-ray. The man sold his pigeons shortly after. There was no testing nor treatment performed on the animals.

### 2.2. Patient 2

A 56-year-old man presented at the emergency department with general malaise, dry cough, headache and fever. He already received amoxicillin for three days from the primary care physician, though symptoms worsened. The man was a truck driver and had an unremarkable medical history but mentioned transporting pigeons on weekends. At the time of presentation, the man was vaccinated once against COVID-19 and had a negative SARS-CoV-2 PCR result. Laboratory tests showed an elevated CRP of 159 mg/L (Reference: <5 mg/L) with a normal leukocyte count. A CT-scan of the chest showed a pneumonic infiltration in the left lower lobe. A diagnosis of pneumonia was made, and the man was empirically treated with clindamycin and levofloxacin for seven days. After five days, the patient had sufficiently improved to be discharged from the hospital. Fourteen days after discharge from the hospital, PCR for *C. psittaci* on sputum returned positive and specific antibodies could be detected as well. The patient was treated for another two weeks with doxycycline (100 mg twice daily). There was no testing or treatment performed on the pigeons.

### 2.3. Patient 3

A 90-year-old woman, double vaccinated against COVID-19 and SARS-CoV-2 PCR-negative, was referred to the emergency department because of presumed upper respiratory tract infection, for which she had received amoxicillin, corticosteroids and aerosol therapy. Nevertheless, she developed more dyspnea, cough and desaturation. She had been living with her son, who kept parrots, though she claimed not to have had close contact with these birds. Laboratory findings included a low CRP of 5 mg/L (reference value < 5 mg/L) though a significant leukocytosis (13.8 × 10^3^/μL, reference value 3.4–9.8 × 10^3^/μL) with a left shift was present. A chest CT scan showed post-infectious changes in the lower lobes. Respiratory multiplex PCR detected parainfluenza virus and sputum showed abundant growth of *Klebsiella pneumoniae* and *Enterobacter cloacae*, for which hospital treatment with levofloxacin was established for five days. *C. psittaci* was detected in the sputum sample by PCR and serology returned positive as well. An additional treatment with doxycycline (100 mg twice daily) was established. The birds (Bourke’s Parrots) tested positive for *C. psittaci* as well and were treated with oxytetracycline.

## 3. Laboratory Investigations

Patient 1 and 2 both showed specific anti-*C. psittaci* IgG antibodies with a titer of 1/400 (reference value <1/100). The third patient had a slightly elevated IgG titer of 1/100 (reference value <1/100). The serum sample of patient 1 was also sent to a second laboratory, which confirmed the positive IgG result (122 U/mL, reference value < 22 U/mL).

The sputum sample of patient 1 was positive by nested *C. psittaci*-specific PCR and culture in Buffalo Green Monkey (BGM) cells. The *C. psittaci ompA* genotyping real-time PCR, directly on sputum, was unsuccessful. However, when using cell culture harvest, the sample could be typed as *ompA* genotype B (Cycle Threshold (CT)-value of 33.95). The sputum sample of patient 2 was positive by nested PCR. Culture was unsuccessful because all monolayers were destroyed by microbial contaminants. Nevertheless, we were able to confirm the nested PCR result as genotyping, performed directly on sputum, revealed the presence of *C. psittaci ompA* genotype B (CT-value of 31.88). The sputum sample of the third patient was positive by nested PCR as well, whereas genotyping was negative for genotypes A to F and E/B. Culture of sputum was negative as well.

From only one patient, the 90-year-old lady, the birds (Bourke’s parrots) were tested by a local veterinarian and confirmed to be infected by *C. psittaci,* but no samples were sent to the national reference center for *C. psittaci* (University of Ghent) for genotyping.

An overview of the laboratory investigations can be found in Table 1.

## 4. Discussion

In this study, we describe three cases of psittacosis, diagnosed during the third and fourth wave of the COVID-19 pandemic in Belgium [10]. A detailed anamnesis showed that all three patients had regular contact with birds and laboratory investigations were initiated to establish a diagnosis. *C. psittaci* infections were confirmed by both PCR and serology and appropriate treatment was initiated to aid the full recovery of the patients. 

While COVID-19 is the most prevalent, if not the most feared, respiratory pathogen among humans since 2019, *C. psittaci* should certainly not be forgotten as a causative agent of pneumonia as well. With humans being the primary host of COVID-19, *C. psittaci* predominantly infects birds and is only occasionally transmitted to humans [11]. The global prevalence of *C. psittaci* in birds is estimated around 20% and its transmission to humans has been reported regularly [12,13]. In 2020, Rybarczyk et al., presented the epidemiological data on psittacosis in Belgium. Since 2010, a small increase in reported cases was observed, but in 2017, the reported cases almost doubled (44 cases) compared to the two previous years. Remarkably, from 2015 to 2017, the mandatory notification system registered only 24% (22 cases in total) of all the cases reported in laboratories [1]. Although two of our three patients were older than 85 years, only 16.3% (39/239) of all infected people in Belgium since 2014 were older than 65 years [14]. Parallel to reports on other diseases, underdiagnosis of psittacosis in elderly can be attributed to limited access to health care, lower levels of education, living in a nursing home and, importantly, to the absence of typical clinical signs [15]. Furthermore, a history of contact with birds should outweigh the importance of a patient’s age in suspecting psittacosis. 

Although an increase in reported cases is observed, the real burden of psittacosis is believed to be underestimated. The reasons for this underestimation are multiple. First of all, psittacosis is a systemic infection which most frequently presents with flu-like symptoms, with highly variable severity and organ involvement. As seen in our three cases, chest imaging and routine laboratory findings are often indistinguishable from other causes of (atypical) pneumonia, including *C. pneumoniae*, *Mycoplasma pneumoniae*, *Legionella pneumoniae*, influenza and SARS-CoV-2 [16]. Second, in community acquired pneumonia (CAP), there is no standard recommendation of extensive testing to find a causative agent and treatment is frequently empirically established. Many patients, therefore, receive the diagnosis of pneumonia without the causative agent being specified [1,17,18]. A study from Raeven et al., performed on 980 patients, concluded that atypical causative agents in CAP are associated with, respectively, the non-respiratory season (May to October), age (<60 years), male gender and absence of COPD. Therefore, testing for atypical agents should be considered in patients younger than 60 years old who are admitted with CAP from early May to early October [17]. Some authors also recommend diagnostic testing when other organisms are ruled out or if there is a history of close contact with birds or other psittacosis cases [6]. Based on our three cases, we want to emphasize the importance of a thorough medical anamnesis, regardless of patient age, as a history of contact with birds plays a key role in setting the diagnosis of psittacosis. 

In this study, diagnosis was performed by a combination of PCR, serology and culture methods. PCR testing is both rapid and sensitive but preferentially requires a sample from the lower respiratory tract (i.e., BAL or sputum) taken within 4 weeks after the onset of symptoms. As the sensitivity of PCR might decrease when the sample is not taken in the acute phase of the infection or when samples are collected from the upper respiratory tract, PCR is often combined with serological tests. Antibody titers can remain positive for over six months, but serological tests are not 100% specific as antibodies can cross-react with other *Chlamydia* species [19,20,21]. A seroconversion or rise in titer is required to confirm the diagnosis of an acute *C. psittaci* infection. However, paired sera from infected people will not always test positive. As an example, we refer to a paper of De Boeck et al. (2016), describing the case of a 54-year-old hospitalized woman who was diagnosed with psittacosis after the purchase of a sick lovebird in a pet shop. Both acute and convalescent blood samples were IgM and IgG negative [22]. False-negative results may be explained by the lack of sensitivity of certain serological tests or the intake of antibiotics 2–3 weeks prior to testing, hampering the development of antibodies [22,23]. Further, individuals carrying genetic mutations in Toll-Like Receptors (TLR), or Nucleotide-binding Oligomerization Domain (NOD) families may fail to recognize the pathogen, resulting in the absence of an adequate immune response. This has already been demonstrated for *C. trachomatis* and might also be applicable to other chlamydiae [24,25]. In spite of possible false-positive and false-negative results, serology can still be of use in outbreak situations, when infections are often discovered after the acute phase and the sensitivity of PCR testing is thereby lowered. Further, a serological test is often available in clinical labs while PCR for *C. psittaci* is sometimes not. Outsourcing of PCR prolongs the time-to-result as additional sample shipping is required and can be avoided by implementing *C. psittaci* in molecular test panels in the clinical labs. 

Both *C. psittaci* DNA and *C. psittaci* IgG antibodies were found for all three patients, revealing an acute infection. However, only the sputum sample from the first patient was also culture positive. Before the arrival of molecular techniques, the culture of *Chlamydiaceae* was the golden standard, as it is the only method to demonstrate the viability of the agent. Nevertheless, culture is difficult, labor-intensive and often requires a biosafety level 3 laboratory given the outbreak potential of the microorganisms and its possible severe disease course. Further, the cultivability of field strains largely varies and is also affected by transport time and temperature, transport medium, prior antibiotics use and the presence of other microorganisms in the sample. It is therefore routinely replaced by other methods [6,26]. As all patients received antibiotics before the sample was collected, it is not surprising that the viability of the agent could only be demonstrated in one sample. 

Finally, genotyping revealed that patient 1 and 2 both were infected with *C. psittaci* genotype B, whereas no genotype could be identified for patient 3. Genotyping can help to map clusters of cases and prove transmission from an animal to a patient, as each genotype has its specific host preferences [6,27]. The genotype-specific PCR developed by Geens et al. (2005) is able to detect the initial seven avian *ompA* genotypes (A-F, E/B) [28]. Genotype A and B are most prevalent in psittacine birds and pigeons, respectively. Genotype C is mostly found in ducks and geese, D in turkeys and F in psittacine birds, as well as turkeys. Genotype E can be found in a wide variety of hosts and genotype E/B was isolated most frequently from ducks. All genotypes are considered capable of infecting humans, with a spectrum of asymptomatic infection to severe pneumoniae and systemic disease [8,23]. The highly virulent genotype A from infected Psittaciformes is predominantly reported in hospitalized patients; but also genotype B, which was found in two of our patients who both were pigeon enthusiasts, is often detected in *C. psittaci* cases [22,29,30,31]. In our third patient, genotyping unfortunately could not confirm any subtype. Possibly, the third patient was infected with one of the newly proposed genotypes (YP84, CPX0308, I, J, Mat116, R54, 6N) which have been found in psittacine birds and wild birds [32,33]. As no genotyping was performed on her resident son Bourke’s parrots, no further conclusions can be drawn. However, we would like to recommend the implementation of a broader genotyping method, including the mammalian genotypes WC and M56 and the newly proposed avian genotypes. Vorimore et al. (2021) recently published a genotyping method based on single-nucleotide polymorphisms able to discriminate between the established genotypes and the less described genotype Mat116 [34].

With the recent advancements in sequencing technologies, traditional approaches are gradually replaced by other molecular alternatives. An example is the use of untargeted metagenomic next-generation sequencing (mNGS), where both host and microbial DNA are sequenced in parallel. As the interpretation of mNGS is complex and the full procedure still needs standardization, it is not routinely used yet [35]. However, this precise and rapid method (average time-to-result is 48 h) has already been applied in diagnosing *C. psittaci* pneumonia [16,36].

As psittacosis is a notifiable disease in Flanders (Belgium), infections must be reported to the Flemish Agency for Care and Health, that subsequently will initiate contact tracing and cluster identification [37]. When an infection can be traced back to contact with a specific bird, this bird must be tested and treated accordingly. However, in this case study, only the birds of the third patient were tested and treated. Both patients 1 and 2 were infected with genotype B, indicating transmission from pigeons, but no pigeons were tested or treated. This non optimal casework, probably due to a lack of time during the COVID crisis, should be avoided because it facilitates the transmission of the pathogen to people and birds. Better communication between laboratories, clinicians and the responsible public agencies has to be encouraged to enable a rapid and effective response.

In conclusion, with this study, we attempted to add to the growing body of evidence that a thorough anamnesis with standard questioning the history of bird contact is indispensable in setting the diagnosis of *C. psittaci*. A history of (frequent) contact with psittacine birds in patients with suggestive though non-specific clinical signs should trigger testing a patient for *C. psittaci* in order to set up the relevant treatment. As a fast diagnosis is often essential for the selection of a suitable treatment, we also recommend the incorporation of *C. psittaci* in syndromic molecular test panels for CAP testing in clinical labs. 

## 5. Materials and Methods

### 5.1. Serology

Blood samples were obtained from all three patients for routine clinical investigations. Samples were taken 7, 5 and 11 days after onset of symptoms for patients 1, 2 and 3, respectively. The sera were sent to an external laboratory (Institute of tropical medicine Antwerp, Belgium) for determination of *C. psittaci*-specific IgG antibodies using a micro-immunofluorescence assay (MIF) from EUROIMMUN (PerkinElmer, Mechelen, Belgium). The serum sample of patient 1 was also sent to another laboratory (Algemeen Medisch Laboratorium Antwerp, Belgium), where they determined *C. psittaci*-specific IgG and IgM antibodies with a MIF from Focus Diagnostics (Cypress, California, USA).

### 5.2. C. psittaci Nested Polymerase Chain Reaction

Sputum samples were collected, respectively, 6, 6 and 11 days after onset of symptoms for patients 1,2 and 3. DNA was extracted from the sputum samples for all patients. For this purpose, the body fluid protocol of the QIAamp DNA mini kit (Qiagen, Antwerp, Belgium) was used. A *C. psittaci*-specific nested PCR was performed on all three DNA extracts, as described by [38]. The PCR detects the outer membrane protein A (*ompA*) gene of all *C. psittaci* genotypes with a sensitivity of 1.0 Inclusion Forming Units (IFU).

### 5.3. Culture

Samples were also used for *Chlamydia* culture. Of each sample, 75 μL was inoculated onto a 24-hour old monolayer of Buffalo Green Monkey (BGM) cells and centrifuged (1200× *g*) for 1 h at 37 °C. Afterwards, monolayers were incubated for six days at 37 °C in complete *Chlamydia* culture medium consisting of Minimal Essential Medium (MEM) (Thermo Fisher Scientific, Geel, Belgium), 5% heat-inactivated fetal calf serum (Greiner Bio-One, Vilvoorde, Belgium), 2% D-glucose (Thermo Fisher Scientific), 2 mM L-glutamine (Thermo Fisher Scientific), 1% MEM vitamins (Thermo Fisher Scientific), 0.1 mg/mL vancomycin (Sandoz NV, Vilvoorde, Belgium), 0.1 mg/mL streptomycin sulphate (Thermo Fisher Scientific, Geel, Belgium), 0.22% cycloheximide (Merck Life Science BV, Overijse, Belgium), 0.25 μg/mL fungizone (Thermo Fisher Scientific) and 0.5 mg/mL gentamycin (Thermo Fisher Scientific). Afterwards, an immunostaining was performed using the IMAGEN™ *Chlamydia* kit (Oxoid™, Geel, Belgium). A sample was considered positive when inclusion-forming units could be identified inside host cells.

### 5.4. Molecular Characterization

DNA extracts of sputum and cell culture were used for *C. psittaci ompA* genotyping. The genotype-specific real-time PCR was performed as described earlier [28]. The assay detects avian *C. psittaci ompA* genotypes A to F and genotype E/B with a sensitivity of 10 copies/μL DNA extract.

## Figures and Tables

**Table 1 pathogens-12-00065-t001:** Overview of the clinical manifestations and laboratory test results of the three patients.

	Patient 1	Patient 2	Patient 3
Clinical manifestations	Dyspnea Fever Malaise	Dry cough Fever Malaise Headache	Cough Dyspnea Desaturation
Analysed samples	Blood and sputum sample	Blood and sputum sample	Blood and sputum
	collected 7 and 6 days after onset of symptoms	collected 5 and 6 days after onset of symptoms	collected 11 days after onset of symptoms
Diagnosis:			
Serology IgG (MIF)	1/400	1/400	1/100
Culture	Positive	Indecisive	Negative
C. psittaci-specific PCR	Positive	Positive	Positive
Genotype-specific PCR	Genotype B	Genotype B	Indecisive

## Data Availability

Not applicable.

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
