# Peer review of "Three Cases of Atypical Pneumonia with Chlamydia psittaci: The Role of Laboratory Vigilance in the Diagnosis of Psittacosis"

_pathogens, 2022, doi:10.3390/pathogens12010065_

Round 1

Reviewer 1 Report

Missault and colleagues present three unrelated case reports of psittacosis following exposure to pigeons or parrots in Belgium. The cases are well documented and reported to increase awareness for a complete diagnostic workup in patients with atypical pneumonia. Very minor comments for consideration.

1.       Case description for patients 1 and 2. Consider adding the specimen type for PCR testing to the narrative. The specimen type is mentioned later in the manuscript but would be helpful if indicated earlier.

2.       Case description for patient 3. There was inconsistent use of italics for bacterial names.

3.       Lines 197-198. How long will a patient remain seropositive after successful treatment?

4.       Line 216. Stating that culture is the gold standard for Chlamydiaceae may be too broad. Molecular testing is the gold standard for Chlamydia trachomatis. And culture for some Chlamydiaceae does not require BSL3 conditions.

Reviewer 2 Report

The manuscript describes three patients with atypical pneumonia who were eventually diagnosed with Chlamydia psittaci after an in-depth anamnesis and laboratory investigation.

My comments:

This is a well-written and balanced manuscript on Chlamydia psittaci infections that can remain undiagnosed and whose prevalence is obviously underestimated.

I have only two major points: 

The first concern is the serology (line 74, table 1 and line 284).

What was the antigen used (purified EB, infected cells, strain of C. psittaci?) Was it a microimmunofluorescence test? 

And the discussion is quite lengthy, most likely it would benefit from condensation.

Minor:

Lines 204-207: Genetic mutations: is data from. C. trachomatis infections only or does it extend to C. psittaci, too?

Reviewer 3 Report

The manuscript is a well-written and informative report on three typical cases of human psittacosis in Belgium during the Covid19 pandemic. I see the merit of the presented work in the fact that it highlights the importance of intensive laboratory diagnosis and precise anamnesis, especially with regard to previous bird contacts of the patients. It shows how by bringing together all the information from the clinic, the laboratory and the veterinary field, a correct diagnosis can be made, which ultimately enables successful therapy and recovery of the patients. On the other hand, the paper does not contain novel information, described cases do not have any particularities, and the reader could question the necessity of publication. Regarding the few data presented, I consider the discussion too long and extensive with many repetitions from previous publications of the group, maybe it can be shortened.

For Chlamydia psittaci PCR detection a rather ancient nested PCR assay was used, while currently a range of Taqman-based real time protocols are available with higher sensitivity and specificity and which are less prone to cross contamination compared to conventional nested protocols. Also the genotyping PCR with its complex system of probes, primers and competition primers with imperfect specificity could be replaced by more straightforward and generally accepted protocols such as MLST or ompA sequencing. Why did you not sequence ompA amplicons from the nested PCR for genotyping?

Some information were missing which possibly could be added:

-      Were all patients hospitalized in the same hospital?

-      Was patient 3 hospitalized at all?

-      Could the immunofluorescence and immunoassays used for serology be specified?

Recommendations for minor changes or additions:

L35/36: a more recent and impressive example of multiple human-to human transmission could be added here:

Zhang Z, Zhou H, Cao H, Ji J, Zhang R, Li W, Guo H, Chen L, Ma C, Cui M, Wang J, Chen H, Ding G, Yan C, Dong L, Holmes EC, Meng L, Hou P, Shi W.

Human-to-human transmission of Chlamydia psittaci in China, 2020: an epidemiological and aetiological investigation.

Lancet Microbe. 2022 Jul;3(7):e512-e520. doi: 10.1016/S2666-5247(22)00064-7. Epub 2022 May 23.

L122: ”,” after as well

L257/258: change “As CAP cases are standard not treated with tetracyclines or macrolides,… “ into “As CAP cases are not treated with tetracyclines or macrolides as standard,… “

L269: add “to” after pathogen

Thank you!
